# Prevalence, types, patterns and risk factors associated with drugs and substances of use and abuse: A cross-sectional study of selected counties in Kenya

**Collins Okoyo**[1,2]*, **Elizabeth Njambi**[2], **Vincent Were**[3], **Sylvie Araka**[1], **Henry Kanyi**[1], **Linnet Ongeri**[4], **Elizabeth Echoka**[5], **Charles Mwandawiro**[1], **Doris Njomo**[1]

**1** Eastern and Southern Africa Centre of International Parasite Control, Kenya Medical Research Institute, Nairobi, Kenya, **2** Department of Data Management and Analysis, Colozzy Data Analytics and Research Solutions, Nairobi, Kenya, **3** Health Economics Research Unit, KEMRI-Wellcome Trust Research Programme, Nairobi, Kenya, **4** Centre for Clinical Research, Kenya Medical Research Institute, Nairobi, Kenya, **5** Centre for Public Health Research, Kenya Medical Research Institute, Nairobi, Kenya

* collinsomondiokoyo@gmail.com

## Abstract

### Background

The increasing trend of drugs and substances abuse (DSA) by different age groups and gender in parts of Kenya is not only a socio-economic problem but a public health concern. There is a need to determine prevalence, types and patterns of DSA by age and gender in order to develop all-inclusive and long-term strategies to prevent and manage the DSA within different communities. In this study we determined the prevalence of DSA, types and patterns of drugs and substances being abused and risk factors associated with this abuse.

### Methods

A descriptive cross-sectional mixed methods study was conducted in four counties; Isiolo, Kajiado, Murang'a and Nyamira, all purposively selected from the 47 counties of Kenya based on the review reports of the Kenya Medical Research Institute's County Cluster Coordinators which indicated that DSA was a priority health concern in the selected counties. From each county, two sub-locations each from two locations in one sub-county were purposively selected. In each sub-location, 225 households were systematically selected. Hence, a total of 3,600 participants were systematically sampled for quantitative data collection using an interviewer-based questionnaire to gather information on magnitude and causes of DSA. Additionally, in each county, qualitative data through in-depth interviews (IDIs) with 16 opinion leaders, 16 healthcare personnel, 16 previous DSAs, at least 5 county personnel, 32 current DSAs; and through 16 focus group discussions (FGDs) were concurrently collected to elicit more information on types, patterns and causes of DSA. The observed overall prevalence of DSA was calculated using binomial logistic regression model and factors associated with DSA analyzed using multilevel logistic regression model. Qualitative data

**Data Availability Statement:** All relevant data are within the article.

**Funding:** The financial support for this research was provided by the Government of Kenya through KEMRI/GRG/15/31. The funders had no role in the study design, data collection and analysis, decision to publish or preparation of the manuscript.

**Competing interests:** The authors have declared that no competing interests exist.

was analyzed using QSR NVIVO version 10, thematically by types, patterns and causes of DSA by age and gender.

## Results

Prevalence of DSA was 86.0% (95%CI: 84.9–87.2) with the highest prevalence being observed in Nyamira County, 89.8% (95%CI: 87.9–91.7). Age-wise, the highest prevalence was observed in persons aged between 45 to 53 years, 89.4% (95%CI: 86.9–92.0), followed by those aged 36 to 44 years, 88.0% (95%CI: 85.4–90.6). Majority of those who abuse drugs and substances were males; 94.5% (95%CI: 93.6–95.4). The most abused drugs or substances were packaged/legal alcohol at 25.2% (745), cigarettes 20.3% (600), local brew (*chang'aa*) 16.3% (482), and khat (*miraa*) 10.5% (311). Risk factors analysis revealed that DSA was significantly higher among males (adjusted odds ratio (aOR) = 7.02 (95%CI: 5.21–9.45), p<0.001), government employees (aOR = 2.27 (95%CI: 1.05–4.91), p = 0.036) and unmarried (aOR = 1.71 (95%CI: 1.06–2.77), p = 0.028).

## Conclusions

These study findings are useful in informing development of specific control programmes which will address age, gender and county needs of DSA in Kenya in order to comprehensively respond to this public health problem. This study was conducted in line with the Kenya National Authority for the Campaign against Alcohol and Drug Abuse (NACADA) mandate to promote use of research on drugs and substances abuse.

## Introduction

Drugs are chemical substances that can change how the body and mind work [1]. They include prescription medicines, over-the-counter medicines, alcohol, tobacco, and illegal drugs, among other substances [2]. Drugs and substances abuse (DSA) can be termed as the habitual use of illegal or legal substances leading to a clinically significant impairment or distress [3]. There is a growing body of literature pointing to the disastrous outcome of DSA globally. According to the World Health Organization (WHO), the global burden of diseases attributable to alcohol and illicit drug use amounts to 5.4% of the total burden of diseases [4]. The World Drug Report of 2018 estimated that a quarter of a billion people, which make up 5.0% of the global adult population, used drugs at least once in 2015 [4]. At the same time, an estimated 29.5 million of those drug users, accounting for 0.6% of the global adult population, suffer from drug use disorders. This means that their drug use is harmful to the point that they may experience drug dependence and require treatment.

In Kenya, several studies have highlighted the serious nature of drugs and substance use [5–7]. The drugs mainly abused in Kenya are either illicit (illegal) such as heroin, cocaine, local brew (*chang'aa*), bhang, kuber, and mandrax, or licit (legal) such as alcohol (beer, wines & spirits), tobacco, and *khat (miraa)*. While use of alcohol and other intoxicating substances is a social behavior which is embedded in communities and cultures, studies have shown that Kenyans generally hold positive attitudes towards consumption of substances such as cigarettes and other nicotine products, packaged liquor, local brew, khat, and other tobacco products [5]. At the same time, trends in drug use are emerging, with amphetamine and related stimulants synthesized in illicit laboratories becoming widely available [3]. The most recent trend is

the diversion, illicit distribution and abuse of prescription drugs that are classified as controlled substances such as synthetic pain medicines, sedative hypnotics and psychostimulants [3], which are medicines that have legitimate use under medical supervision but whose use can quickly become problematic if used inappropriately. The burden of DSA is increasing in Kenya, with current statistics indicating that adolescents and young people aged 10–19 years account for more than 50% of drug users [8]. The negative impact of DSA is reflected in the immediate and long-term effects to the individuals and families concerned as well as the entire society.

If DSA is left unaddressed, the country risks losing generations as well as lagging in development owing to the diversion of resources to address the problem. The government has acknowledged this problem and put in place legislative, social, educational and healthcare management measures to manage and control the abuse [8]. The self-perpetuating nature of DSA problems makes control difficult pointing to a need for a specific control programme which will address age, gender and county needs and respond to the problem comprehensively. Efforts to design and implement workable DSA prevention and control programmes which are embedded in the health systems framework should be intensified. Furthermore, in the devolved government, the Kenya Health Policy 2012–2030 which is aligned to Kenya's vision 2030, the Constitution of Kenya and global health commitments (e.g. the Sustainable Development Goals) provides guidance to the health sector on identification and outlining of the requisite activities in achieving the government's health goals [3]. Therefore, this study sought to outline the prevalence, types, patterns and risk factors associated with DSA in Kenya in order to inform control strategies to be embedded in the health systems framework of Kenya for responding to gender and age needs in prevention and management of DSA. The study also sought to generate knowledge that may inform development of county specific response to DSA in Kenya.

## Materials and methods

### Study design

The study used a descriptive cross-sectional design which employed mixed methods data collection technique where quantitative and qualitative data were collected concurrently. Quantitative data was collected from the household heads to determine the prevalence, types, patterns, and causes of DSA. Qualitative data was collected using in-depth interviews (IDIs) with opinion leaders, healthcare personnel, persons who are currently and those who have previously abused drugs and substances and county leaders. Additionally, focus group discussions (FGDs) were conducted with community groups categorized by age and gender.

### Sample size determination and sampling

Sample size was calculated for the quantitative arm of the study targeting households where at least one person has abused any drugs or substances in the last five years. For each household head, we asked about type of drugs being abused, sources of these drugs, and information about members of the family abusing any type of drugs and substances. The sample size was determined using formula described by Fischer [9] using proportion of households with a least one person who has abused drugs or substances, a margin error of 3.5% and 15% non-response rates;

$$n = \frac{(Z^2)pq}{e^2}$$

Where n is the required sample size, z is the score for a 5% type 1 error for a normal distribution (Z = 1.96), p is the proportion of drugs and substance abusers among the households in the study areas (assumed to be 50%), q = 1-p (50%) and e is the margin of error set at 3.5%. The minimum sample size was 783 households per county. An additional 15% non-response rate was added to raise the sample size to 900 per county. Since the study was conducted in four counties, the total sample size was 3,600 households.

Additionally, qualitative data through IDIs and FGDs were collected from various categories of participants to elicit information on types, patterns, causes, strategies and intervention mechanisms for responding to age and gender needs in prevention and management that should be embedded in the health systems framework. In each county, separate IDIs were conducted with 16 opinion leaders, 16 healthcare personnel, 16 previous DSAs, five to seven county leaders all purposively selected and 32 (16 adults and 16 mature minors) current DSAs were sampled using the snowball method. In addition, 16 FGDs with community groups categorized by age and gender were conducted.

## Study sites and study outline

Given the regional variations in DSA in Kenya, four counties (Isiolo, Kajiado, Murang'a and Nyamira) from the entire 47 counties were purposively sampled for this study (Fig 1). The selection of the four counties was based on review reports of the Kenya Medical Research

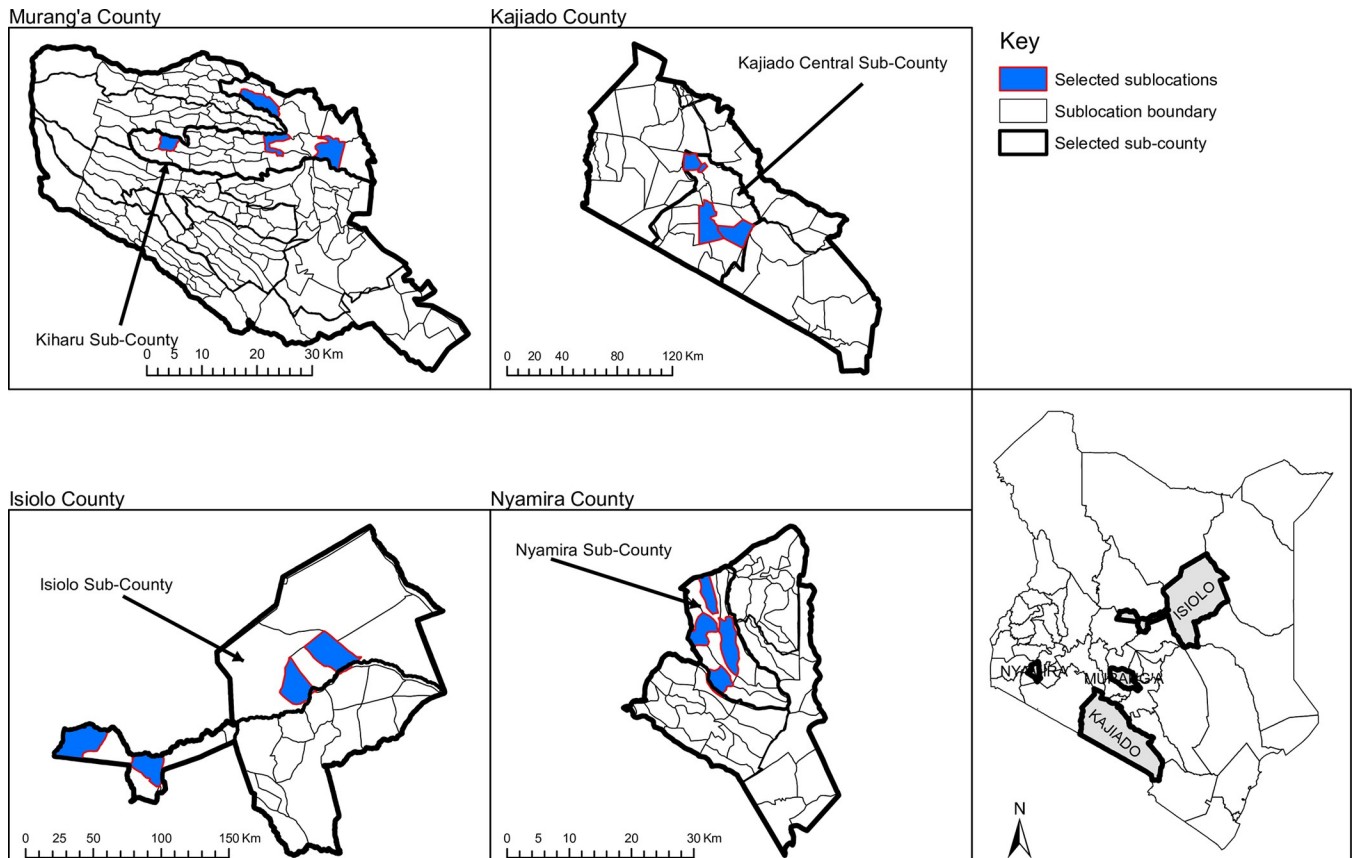

**Fig 1. A map of Kenya showing the sampled study areas (sub-locations).** The map for the study area was created using ArcGIS Desktop version 10.2.2 software (Environmental Systems Research Institute Inc., Redlands, CA, USA). Copyright © Esri. All rights reserved.

Institute (KEMRI) County Cluster Coordinators which indicated that DSA was top on the list of priority health concerns of the selected counties.

As illustrated above, from each sampled county, one sub-county was purposively sampled; subsequently two locations were purposively sampled from the selected sub-county and further two sub-locations were selected from the identified locations. From each sub-location sampled, 225 households were sampled using systematic sampling technique. Starting from a given point, every $k^{th}$ household in each sub-location was systematically selected using fixed, periodic interval obtained as $\left( N_h / n_h \right)^{th}$ household, where $N_h$ is the total number of households in a sub-location and $n_h$ is the calculated sample size of households in a particular sub-location. If a selected household did not have a member who had in the recent past (i.e. past five years) been abusing drugs and substances, we selected the next eligible household. Study areas were varied based on rural and urban settings. Fig 2 shows the study outline schema.

In Murang'a County, Kiharu Sub-County was purposively selected and 949 households from four sub-locations: Gakuyu (254); Karuri (200); Kambirwa (232) and Gikandu (263) were systematically sampled.

In Kajiado County, Kajiado Central Sub-County was purposively selected and 753 households from four sub-locations: Bissil (217); Lenkishon (175); Loormongi (149) and Majengo Township (212) were systematically sampled.

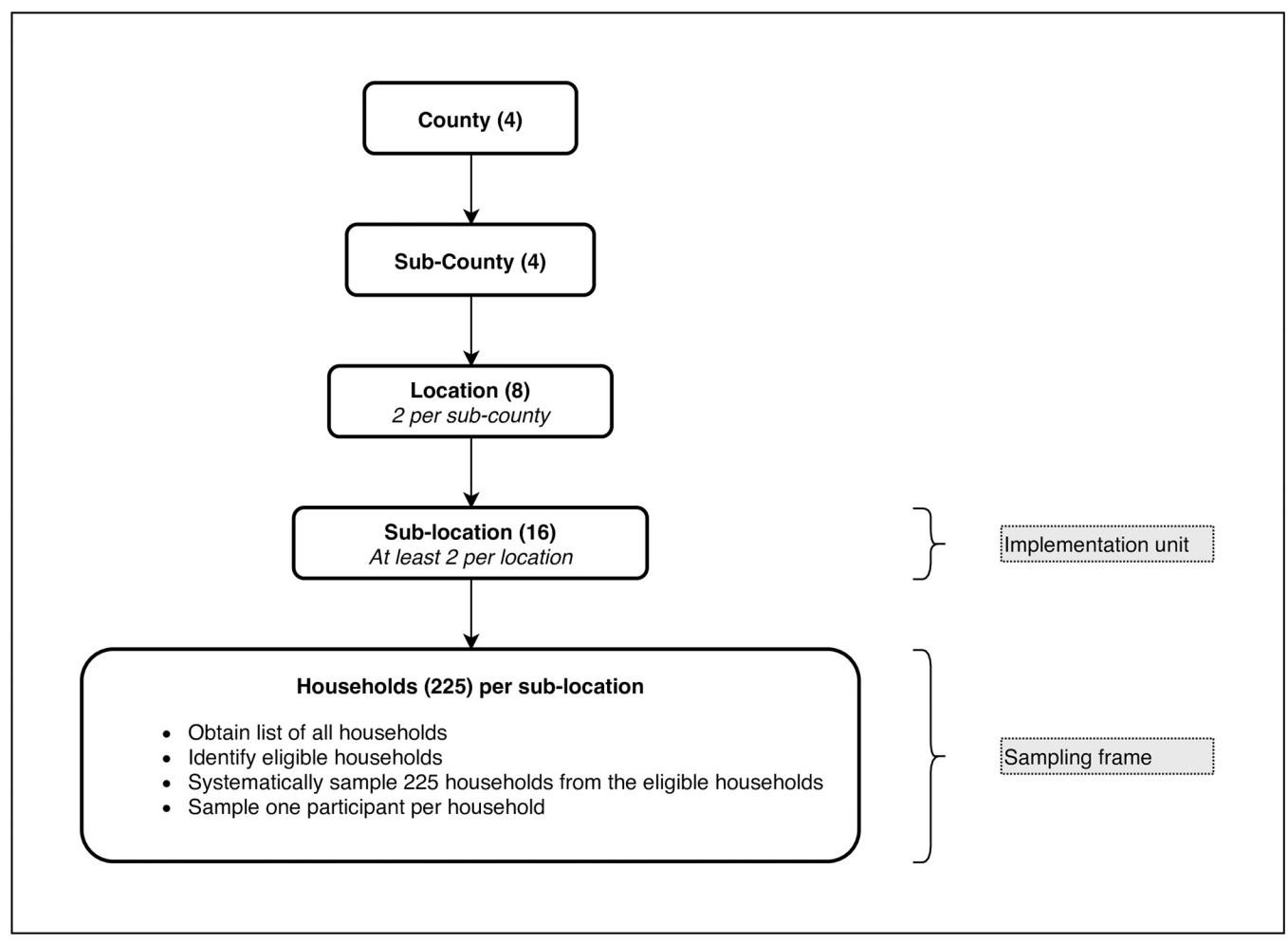

**Fig 2. Study profile.**

In Isiolo County, Isiolo Sub-County was purposively selected and 716 households from four sub-locations: Tulluroba (208); Wabera (162); Bullapesa (155) and Kambi Odha (191) were systematically sampled.

In Nyamira County, Nyamira Sub-County was purposively selected and 1021 households from four sub-locations: Bomanyanya (233); Township (287); Siamani (257) and Miruka (244) were systematically sampled.

## Study population

For the quantitative arm of the study, community members who were household heads from sampled households (adult and youth, both men and women) were targeted to generate information on types, patterns and causes of DSA. For the qualitative arm, IDIs were conducted with a total of 125 current drug users (both mature minors (63) and adults (62) recruited through snowball sampling method and a total of 64 previous drugs and substance abusers purposively selected to generate information on types of drugs and substances abused, patterns and causes of abuse, experiences and intervention mechanisms for prevention and management of abuse. Sixty four (64) IDIs were also conducted with opinion leaders (community, religious and social group) and with twenty one (21) county leaders and sixty four (64) with healthcare providers to elicit information on prevention and management intervention mechanisms. Additionally, sixty four (64) focus group discussions were conducted with community groups stratified by age and gender to elicit more information on causes and patterns of abuse and intervention mechanisms for prevention and management.

For both qualitative and quantitative study arms, participants aged 18 years and above and are current or previous drug or substance abusers, or in households where there is current or previous abuser were included in the study after providing informed consent. In some instances, mature minors (13 to 17 years) who are drug and substance abusers were also included in the study after providing informed assent and getting informed consent from their parents/guardians. Additionally, county, community, religious, social and opinion leaders of the sampled study sites and who are 18 years and above were included in the study. A sample frame of all households in the sub-location was obtained from community registers and households were then systematically selected from the list.

## Ethical approval and consent to participate

The study protocol received ethical approval from KEMRI's Scientific and Ethics Review Unit (SERU No. 3237). Additional approval was provided by the county-level health and security authorities after they were appropriately briefed about the study. At sub-location level, local administration (sub-chief) provided approval, while at household level, the household head provided written consent.

## Data collection, management and analysis

Quantitative and qualitative data were collected separately but concurrently, in each county, to allow for comparison after analysis. Data were collected in each county as follows; November 2016 for Kajiado County, May 2017 for Isiolo County, November 2017 for Murang'a County and June-August 2018 for Nyamira County.

For quantitative data, participants' responses were captured electronically into the Open Data Kit (ODK) system [10], which included in-built data quality checks to prevent data entry errors. Prevalence of DSA was defined as the proportion of participants who currently (i.e. at the period of the survey) abuse any type of drug or substance. Observed overall prevalence was calculated at county level and 95% confidence intervals (CIs) obtained using binomial logistic

regression model. For the purpose of this analysis, the following age categories were used: ≤26, 27–35, 36–44, 45–54, 55–63, 64–72 and >72 years. Factors associated with DSA were first analyzed using univariable analysis and described as odds ratio (OR) using mixed effects logistic regression model. In multivariable analysis, adjusted OR (aOR), were obtained by mutually adjusting all minimum generated variables using multivariable mixed effects logistic regression model at 95% CI. All quantitative analyses including graphs were carried out using STATA version 14.0 (STATA Corporation, College Station, TX, US). The map for the study area was created using ArcGIS Desktop version 10.2.2 software (Environmental Systems Research Institute Inc., Redlands, CA, USA) [11].

For the qualitative arm, the data collection was moderated by two social scientists from KEMRI assisted by 32 trained field assistants with the aid of interview guides and FGD guides using the local language. The design was iterative and there was a back and forth process which included data collection and analysis and further sample selection therefore giving early insights and influencing selection of more participants up to the point where no newer information was being gathered. Standard procedures including maintaining a neutral stance, probing and allowing the respondents to express themselves without asking leading questions, asking general questions before specific questions and varying questions wording to avoid seeming repetitive were adhered to [12]. Each FGD and each IDI took a minimum of 40 minutes to a maximum of 60 minutes and both were held in private areas to ensure participants' confidentiality. Notes were taken during the data collection process and voice recorders used to record all the information in the local language.

The recorded data were coded and later transcribed and translated into English. Double transcription and translation and back translation was done among the investigators so as to agree on the meaning of the transcripts and minimize bias. The hard copies of the data were stored in lockable and secure cabinets. To ensure quality control, the soft copies were stored in computers with passwords, with authorized access by the Principal Investigator to the study investigators. The data were coded and entered into QSR NVIVO version 10 for management and analysis. Manual analysis was further conducted according to study themes which were determined prior to the analyses. A code sheet was created following the IDIs and FGDs guides after which, the textual data was coded into selected themes and a master sheet analysis was carried out, giving all the responses a theme. Thematic analysis was used where responses were categorized into themes and then ideas formulated by looking at the patterns of responses [13]. The analyzed data were presented in text form. Representative quotes were embedded within the results to illustrate themes, with minor grammatical alterations to improve readability.

## Results

### Socio-demographic and socio-economic characteristics of the quantitative arm of the study participants

A total of 3,439 participants were sampled for the quantitative arm of the study. This comprised of 1,021 participants (29.7%) in Nyamira County, 949 (27.6%) in Murang'a County, 753 (21.9%) in Kajiado County and 716 (20.8%) in Isiolo County. The mean age of the participants was 42.3 years (standard deviation (SD) 15.6 years; range 18–93 years). Majority of the participants were males 68.2% (2,345) and 31.8% (1,093) were females. Most of them were married at the time of the study 63.4% (1,704), while 18.5% (496) were single, 10.8% (289) were widowed and 7.3% (197) were divorced. The highest level of formal education for most participants was secondary level 22.2% (596) while 6.8% (183) had never been to school. Participants of Christian belief were the majority 77.5% (2,082), Islam belief were 16.8% (450) and those who were non-practicing were 5.5% (148) (Table 1).

**Table 1. Overall prevalence (%) of any drug and substance abuse.**

| Demographics | Household heads | Prevalence of any drug or substance |
|---|---|---|
| | (n = 3,439) | % (95% CI); n |
| **Overall** | **3,439** | **86.0% (95%CI: 84.9–87.2); n = 2,957** |
| County | | |
| Kajiado | 753 (21.9%) | 79.5% (95%CI: 76.7–82.5); n = 598 |
| Murang'a | 949 (27.6%) | 85.5% (95%CI: 83.2–87.7); n = 811 |
| Nyamira | 1,021 (29.7%) | 89.8% (95%CI: 87.9–91.7); n = 917 |
| Isiolo | 716 (20.8%) | 88.1% (95%CI: 85.8–90.5); n = 631 |
| Age in years | | |
| 18–26 | 538 (15.7%) | 86.9% (95%CI: 84.2–89.9); n = 468 |
| 27–35 | 895 (26.1%) | 87.9% (95%CI: 85.8–90.1); n = 787 |
| 36–44 | 600 (17.5%) | 88.0% (95%CI: 85.4–90.6); n = 528 |
| 45–53 | 568 (16.5%) | 89.4% (95%CI: 86.9–92.0); n = 508 |
| 54–62 | 405 (11.8%) | 85.4% (95%CI: 82.1–88.9); n = 346 |
| 63–71 | 269 (7.8%) | 72.9% (95%CI: 67.7–78.4); n = 196 |
| > 72 | 159 (4.6%) | 75.5% (95%CI: 69.1–82.5); n = 120 |
| Gender | | |
| Male | 2,345 (68.2%) | 94.5% (95%CI: 93.6–95.4); n = 2,216 |
| Female | 1,093 (31.8%) | 67.8% (95%CI: 65.1–70.6); n = 741 |
| Marital status | | |
| Single | 496 (18.5%) | 93.9% (95%CI: 91.9–96.1); n = 466 |
| Currently married | 1,704 (63.4%) | 88.4% (95%CI: 86.9–89.9); n = 1,507 |
| Divorced | 197 (7.3%) | 89.3% (95%CI: 85.1–93.8); n = 176 |
| Widow/widower | 289 (10.8%) | 72.7% (95%CI: 67.7–77.9); n = 210 |
| Education level | | |
| None | 183 (6.8%) | 81.9% (95%CI: 76.6–87.7); n = 150 |
| Primary | 1,116 (41.6%) | 83.5% (95%CI: 81.4–85.7); n = 932 |
| Secondary | 1,001 (37.3%) | 91.4% (95%CI: 89.7–93.2); n = 915 |
| College | 386 (14.4%) | 93.8% (95%CI: 91.4–96.2); n = 362 |
| Occupation | | |
| Subsistence farmer | 860 (32.0%) | 78.9% (95%CI: 76.3–81.7), n = 679 |
| Government employee | 195 (7.3%) | 94.9% (95%: 91.8–98.0), n = 185 |
| Student | 125 (4.7%) | 93.6% (95%CI: 89.4–97.9), n = 117 |
| Large scale farmer | 13 (0.5%) | 84.6% (95%CI: 67.1–106.7), n = 11 |
| Private worker | 141 (5.3%) | 93.6% (95%CI: 89.7–97.7), n = 132 |
| Casual laborer | 654 (24.4%) | 93.6% (95%CI: 91.7–95.5), n = 612 |
| Self employed | 584 (21.7%) | 91.9% (95%CI: 89.8–94.2), n = 537 |
| Others | 114 (4.2%) | 75.4% (95%CI: 67.9–83.8), n = 86 |
| Dwelling | | |
| Permanent | 1,182 (44.0%) | 88.4% (95%CI: 86.6–90.3); n = 1,045 |
| Semi-permanent | 1,018 (37.9%) | 85.8% (95%CI: 83.6–87.9); n = 873 |
| Temporary | 486 (18.1%) | 90.7% (95%CI: 88.2–93.4); n = 441 |

The main occupation reported was subsistence farming 32.0% (860), followed by casual labor 24.4% (654), self-employment 21.7% (584), government employment 7.3% (195), private worker 5.3% (141), student 4.7% (125), and large-scale faming 0.5% (13). Most participants resided in a permanent dwelling, 44.0% (1,182), while 18.1% (486) in a temporary dwelling. The main sources of water for the households were piped water into either dwelling 18.7%

(503) or yard 18.5% (498). However, 17.6% (473) used surface water as their main source. More than half of the households were using traditional pit latrines 56.4% (1,515) followed by ventilated improved pit latrines 32.4% (871), and flush toilet 9.1% (245). However, 1.6% (42) of the households did not have any toilet facility. Majority of the households, 88.9% (952), were covered by a health insurance with majority (91.8%) reporting that they were covered by the National Health Insurance Fund (Table 1).

## Socio-demographic characteristics of the qualitative arm of the study participants

In Isiolo County, majority of the participants in all the categories of the IDIs were male. The county leaders comprised of one assistant county commissioner, one county probation officer, one medical superintendent, one county prisons commander and one police commander. The mean age for IDI category was 47 years. The community opinion leaders included police officers, teachers, chairlady of women groups, Imams, and village elders. The mean age for community opinion leaders category was 37 years. Regarding the health workers, five were nurses, six clinical officers, three public health officers, one community health extension worker and one social worker. Additionally, sixteen (16) FGDs were conducted with 187 participants categorized by age and gender. Slightly over one half of the participants in the FGDs were primary school leavers. A majority of them were casual labourers and of the Muslim faith.

In Kajiado County, majority of those who participated in the various IDIs categories were male. A total of 16 IDIs were conducted with health care workers; nurses, clinical officers, doctors, and public health officers. Seven county leaders participated and they included county commissioner, officer commanding police division, chief health officer, county official, quality assurance officer, senior nursing officer and the social development officer. Sixteen opinion leaders included pastors, chiefs, teachers and village elders, among whom 12 were males and 4 females and all were married. Additionally, 31 current abusers also participated, among whom 16 were adults and 15 mature minors. Further, a total of 190 participants categorized by age and gender participated in 16 FGDs. The mean age of the participants was 31 years and a majority had secondary school level of education and above.

In Murang'a County, four county leaders (two assistant county commissioners, county public health officer and a chief county public health officer), 16 healthcare workers, 16 opinion leaders (teachers, chairs of social and economic groups, religious leaders, village elders), 32 (16 adults and 16 mature minors) current DSAs and 16 previous DSAs participated in IDIs. Additionally, a total of 191 participants took part in 16 FGDs. The gender representation was 50.5% male and 49.5% female and a large majority were Christians.

In Nyamira County, a total of 80 participants took part in the various categories of IDIs. A large majority (78.8%) were male and all were Christians. The mean age for the opinion leaders was 45 years, all were married and majority had attained post primary level of education. Of the 16 participants who had previously abused drugs and substances, nearly all 15/16 were males and 10 had post primary education. Of the 32 (16 adults and 16 mature minors) who were current drug and substance abusers interviewed, a large majority (82%) were male, and more than a half had secondary level of education. Of 16 healthcare workers interviewed, 8 were nursing officers, 2 clinical officers, one public health officer, 2 pharmacists, one social worker, one doctor and one laboratory technologist. A total of five county leaders were interviewed; County Director of Health, County Health Promotion Officer, County Mental Health Coordinator, Assistant County Commissioner and the County Public Health Officer. Additionally, a total of 191 participants with a median age of 38 years took part in 16 FGDs. The gender representation was 49.7% male and 50.3% female. A large majority (92.1%) were Christians and their main occupation was farming.

## Prevalence of drugs and substances abuse

Overall, 86.0% (95%CI: 84.9–87.2) of the participants abuse any type of drug or substance. Among the counties, the highest prevalence was observed in Nyamira 89.8% (95%CI: 87.9–91.7), followed by Isiolo 88.1% (95%CI: 85.8–90.5), Murang'a 85.5% (95%CI: 83.2–87.7) and Kajiado 79.5% (95%CI: 76.7–82.5) (Table 1).

Among the participants, the highest prevalence was observed in those aged between 45 to 53 years, 89.4% (95%CI: 86.9–92.0), and lowest prevalence in those above 72 years, 75.5% (95%CI: 69.1–82.5). Majority of those who abuse drugs and substances were males as compared to females, 94.5% (95%CI: 93.6–95.4) and 67.8% (95%CI: 65.1–70.60) respectively (Table 1).

Household heads were additionally asked questions on behalf of other members of their household who use any drug or substance. Overall, 50.6% of the household heads had members who abuse any type of drugs or substances. The reported age range of these household members was 14 to 110 years with mean age of 34.7 years (SD = 13.5 years). In each household, between one to seven members reportedly used any drug or substance.

## Types, sources and patterns of drugs and substances abuse

Overall, the most abused drugs and substances were packaged/legal alcohol (25.2%; 745), cigarettes (20.3%; 600), *chang'aa* (16.3%; 482), *miraa* (10.5%; 311), tobacco (6.7%; 198), prescription drugs (5.5%; 163), bhang' (5.2%; 155), and *muguka* (4.9%; 147) (Table 2, Fig 3).

According to the surveyed counties, packaged alcohol and cigarettes were the most abused drugs in Kajiado and Murang'a counties. *Chang'aa* and packaged alcohol were the most abused in Nyamira, while *miraa* and cigarettes were the most abused in Isiolo. In all the

**Table 2. Types of drugs and substances abused by respondents according to age group.**

| Types of drugs and substances abused | Number surveyed; | Age group (in years) | | | | | | |
|---|---|---|---|---|---|---|---|---|
| | N = 3,439 | 18–26; n = 538 | 27–35; n = 895 | 36–44; n = 600 | 45–53; n = 568 | 54–62; n = 405 | 63–71; n = 269 | >72; n = 159 |
| | n (%) | | | | | | | |
| Cigarettes | 600 (20.3%) | 58 (12.4%) | 155 (19.7%) | 142 (26.7%) | 112 (22.1%) | 75 (21.7%) | 31 (15.7%) | 26 (21.7%) |
| Snuff/chewed/piped tobacco | 198 (6.7%) | 12 (2.6%) | 27 (3.4%) | 21 (3.9%) | 35 (6.9%) | 46 (13.3%) | 28 (14.1%) | 28 (23.3%) |
| Kuber | 35 (1.2%) | 4 (0.9%) | 15 (1.9%) | 6 (1.1%) | 8 (1.6%) | 2 (0.6%) | - | - |
| Shisha | 49 (1.7%) | 24 (5.1%) | 17 (2.2%) | 5 (0.9%) | 2 (0.4%) | 1 (0.3%) | - | - |
| Packaged/legal alcohol | 745 (25.2%) | 122 (26.1%) | 207 (26.3%) | 126 (23.7%) | 138 (27.2%) | 86 (24.9%) | 40 (20.2%) | 25 (20.8%) |
| Chang'aa | 482 (16.3%) | 48 (10.3%) | 108 (13.7%) | 87 (16.4%) | 105 (20.7%) | 60 (17.4%) | 47 (23.7%) | 26 (21.7%) |
| Bhang | 155 (5.2%) | 50 (10.7%) | 49 (6.2%) | 27 (5.1%) | 13 (2.6%) | 8 (2.3%) | 8 (4.0%) | - |
| Miraa | 311 (10.5%) | 77 (16.5%) | 108 (13.7%) | 65 (12.2%) | 31 (6.1%) | 23 (6.7%) | 6 (3.0%) | 1 (0.8%) |
| Muguka | 147 (4.9%) | 40 (8.6%) | 67 (8.5%) | 26 (4.9%) | 9 (1.8%) | 4 (1.2%) | 1 (0.5%) | - |
| Heroine/brown sugar | 2 (0.1%) | 2 (0.4%) | - | - | - | - | - | - |
| Cocaine coke/crack | 5 (0.2%) | - | 2 (0.3%) | - | 1 (0.2%) | 1 (0.3%) | 1 (0.5%) | - |
| Petroleum/paints/thinner/glue | 2 (0.1%) | 2 (0.4%) | - | - | - | - | - | - |
| Prescription drugs | 163 (5.5%) | 16 (3.4%) | 23 (2.9%) | 18 (3.4%) | 32 (6.3%) | 32 (9.3%) | 33 (16.7%) | 9 (7.5%) |
| Sedatives or sleeping pills | 46 (1.6%) | 5 (1.1%) | 8 (1.0%) | 6 (1.1%) | 15 (2.9%) | 5 (1.5%) | 2 (1.0%) | 5 (4.2%) |
| Morphine, codeine pethidine | 3 (0.1%) | 2 (0.4%) | - | - | - | - | 1 (0.5%) | - |
| Amphetamine type stimulants | 13 (0.4%) | 3 (0.6%) | 2 (0.3%) | 1 (0.2%) | 6 (1.2%) | 1 (0.3%) | - | - |
| Hallucinogens | 4 (0.1%) | 2 (0.4%) | - | 1 (0.2%) | - | 1 (0.3%) | - | - |
| Mandrax | 1 (0.0%) | - | - | - | 1 (0.2%) | - | - | - |

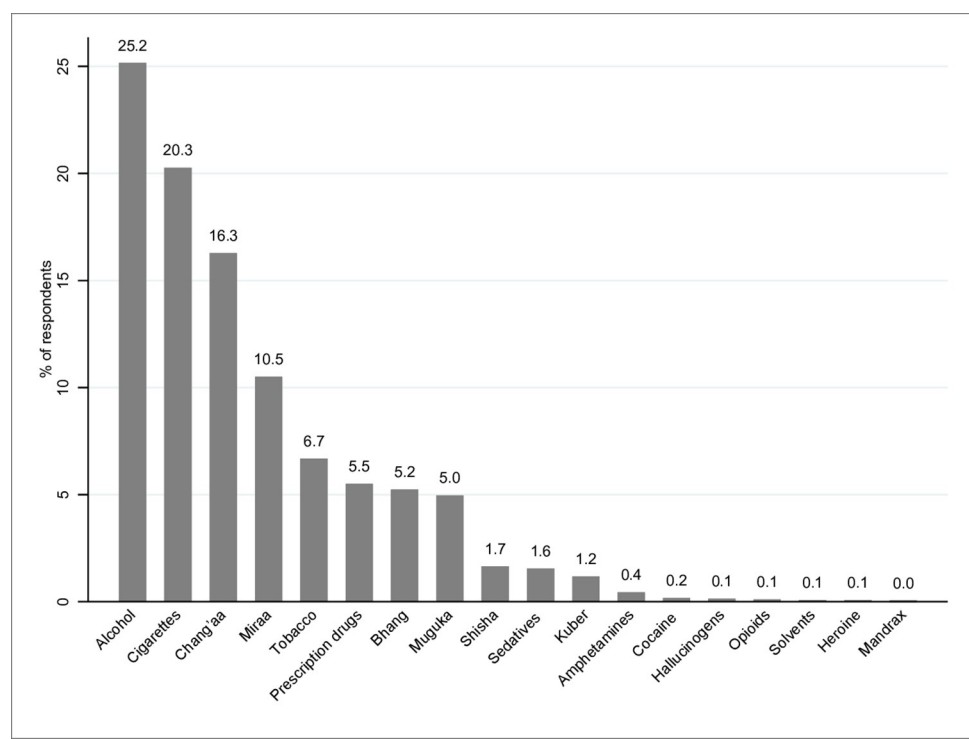

**Fig 3. Overall types of drugs and substances abused by the study participants.**

**Table 3. Types of drugs and substances abused by respondents according to county and gender.**

| Types of drugs and substances abused | Frequency; N = 3,439 | County | | | | Gender | |
|---|---|---|---|---|---|---|---|
| | | Kajiado; | Murang'a; | Nyamira; | Isiolo; | Male; | Female; |
| | n (%) | n = 753 | n = 949 | n = 1,021 | n = 716 | n = 2,345 | n = 1,093 |
| Cigarettes | 600 (20.3%) | 132 (21.9%) | 193 (23.7%) | 139 (15.2%) | 136 (21.6%) | 511 (23.1%) | 89 (11.9%) |
| Snuff/chewed/piped tobacco | 198 (6.7%) | 64 (10.6%) | 62 (7.6%) | 28 (3.1%) | 44 (6.9%) | 121 (5.5%) | 77 (10.4%) |
| Kuber | 35 (1.2%) | 2 (0.3%) | 7 (0.9%) | 20 (2.2%) | 6 (0.9%) | 27 (1.2%) | 8 (1.1%) |
| Shisha | 49 (1.7%) | 5 (0.8%) | 17 (2.1%) | 2 (0.2%) | 25 (3.9%) | 21 (0.9%) | 28 (3.8%) |
| Packaged/legal alcohol | 745 (25.2%) | 206 (34.1%) | 268 (32.9%) | 204 (22.4%) | 67 (10.6%) | 600 (27.1%) | 145 (19.5%) |
| Chang'aa | 482 (16.3%) | 68 (11.3%) | 25 (3.1%) | 339 (37.2%) | 50 (7.9%) | 350 (15.8%) | 132 (17.7%) |
| Bhang | 155 (5.2%) | 23 (3.8%) | 63 (7.7%) | 28 (3.1%) | 41 (6.5%) | 130 (5.9%) | 25 (3.4%) |
| Miraa | 311 (10.5%) | 49 (8.1%) | 26 (3.2%) | 35 (3.8%) | 201 (31.9%) | 211 (9.5%) | 100 (13.4%) |
| Muguka | 147 (4.9%) | 46 (7.6%) | 54 (6.6%) | 4 (0.4%) | 43 (6.8%) | 119 (5.4%) | 28 (3.8%) |
| Heroine/Brown sugar | 2 (0.1%) | 1 (0.2%) | 1 (0.1%) | - | - | 2 (0.1%) | - |
| Cocaine coke, crack | 5 (0.2%) | 1 (0.2%) | - | - | 1 (0.2%) | 4 (0.2%) | 1 (0.1%) |
| Solvents(petroleum/paints/thinner/glue) | 2 (0.1%) | - | - | 1 (0.1%) | 1 (0.2%) | 1 (0.1%) | 1 (0.1%) |
| Prescription drugs | 163 (5.5%) | 7 (1.2%) | 64 (7.9%) | 85 (9.3%) | 7 (1.1%) | 95 (4.3%) | 68 (9.1%) |
| Sedatives or sleeping pills | 46 (1.6%) | - | 26 (3.2%) | 14 (1.5%) | 6 (0.9%) | 18 (0.8%) | 28 (3.8%) |
| Opioids (morphine, codeine pethidine) | 3 (0.1%) | - | 3 (0.4%) | - | - | 1 (0.1%) | 2 (0.3%) |
| Amphetamine type stimulants | 13 (0.4%) | - | 5 (0.6%) | 5 (0.6%) | 3 (0.5%) | 4 (0.2%) | 9 (1.2%) |
| Hallucinogens | 4 (0.1%) | - | - | 4 (0.4%) | - | 1 (0.1%) | 3 (0.4%) |
| Mandrax | 1 (0.0%) | - | - | 1 (0.1%) | - | 1 (0.1%) | - |

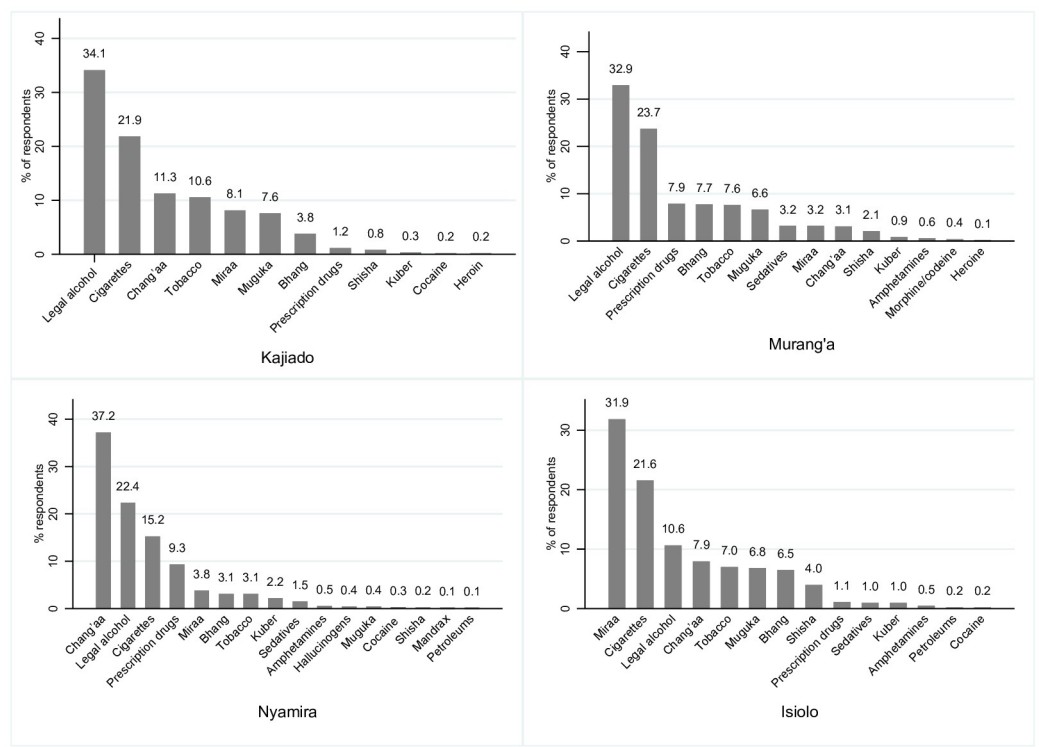

**Fig 4. Types of drugs and substances abused by the study participants grouped by the surveyed counties.**

counties, hard drugs like cocaine, heroin, mandrax and morphine were least abused (Table 3, Fig 4).

Results from the qualitative survey show that in Isiolo County the most frequently abused drugs and substances according to the results from the IDIs with the current abusers as well as with the opinion leaders and FGDs participants included bhang, miraa, glue, alcohol, *chang'aa*, tobacco and cigarettes.

> "*In this area most people take a lot of chang'aa and it's a major problem, people smoke bhang. Cigarette smoking is practiced in this community. Members of the community chew a lot of muguka and wine. Youths use tobacco. Glue and gum is another substance*" (FGD- male youth).

> "*The drugs and substances that I know of in this community are miraa, cigarettes and alcohol*" (IDI- opinion leader).

> "I *am one of the users so it is there in this community; cigarettes, bhang, muguka, miraa, shisha, alcohol, chang'aa and petroleum*" (IDI- current adult drugs and substance abuser).

In Kajiado County, results from the FGD participants and the IDIs among the opinion leaders and current drugs and substance abusers, showed that the most abused drugs and substances included alcohol, bhang, tobacco and cigarettes.

> "*Yes, they abuse bhang, chang'aa, shisha, second generation, tobacco, cigarettes.*" (IDI-Community opinion leader).

Results from the qualitative survey with opinion leaders in Murang'a County highlighted the abuse of prescription drugs as a problem in the county.

*"In fact those users get some drugs from the hospital especially those from the mental department because those are drugs abused and most are given through back door. Thus for those who need them they end up lacking at their time of need. Some drugs abused are even bought from the chemist."* (IDI-opinion leader).

For Nyamira County the qualitative results showed that the abused drugs and substance are *chang'aa*, bhang, kuber, alcohol, cigarettes, tobacco, *busaa*, *miraa*, cocaine and shisha.

*"Chang'aa and bhang, I use when I have money if I don't have money I don't use."* (IDI- Current drugs and substance abuser, mature minor)

*"I'm a heavy drinker; I smoke bhang and any drink that I come across."* (IDI- current drugs and substance abuser, adult)

The opinion leaders also reiterated that the following drugs were being abused in the community; chang'aa, cigarettes, kuber, bhang, tobacco. The youth of both genders were reported to be the main abusers, some of whom are students in school. The qualitative reports also indicated that more males abused drugs and substances compared to females.

*"In this community we have a lot of drugs that are being abused like bhang, kuber, cigarettes, alcohol."* (IDI-Opinion leader).

By gender, packaged alcohol was the most abused by males (27.1%; 600), followed by cigarettes (23.1%; 511) and chang'aa (15.8%; 350). Similarly, among females, packaged alcohol was the most abused (19.5%; 145), followed by chang'aa (17.7%; 132) and miraa (13.4%; 100) (Table 3, Fig 5).

The qualitative survey results of the IDIs with current drug abusers, with opinion leaders and FGDs participants in Isiolo County showed that most drugs and substances were abused by male youths especially during school holidays, weekends and during community festivals. The results further revealed that females were less likely to abuse the drugs and substances because of the cultural norm which discourages the use of these substances by females.

*"I get cravings and urge always on daily basis. The problem is when I do not get the cash to purchase them. On such days you do not use these drugs only because I don't have the money to buy them"* (IDI- current drugs and substance abuser, male adult).

*"Like miraa I chew on Saturdays only but cigarettes I take it daily and for chang'aa every day"* (IDI- current drugs and substance abuser, male mature minor).

Similarly, in Kajiado County findings from the FGD participants and the IDIs among the opinion leaders and current abusers, drugs and substances abuse was more common among the males both youth and adults than among females. Furthermore, the interviews revealed that female was less likely to abuse the drugs and substances because of the cultural norm (mostly from the Maasai community) which discourage the use of these substances.

*"Yes, they abuse bhang, chang'aa, shisha, second generation, tobacco, cigarettes."* (IDI-Community opinion leader)

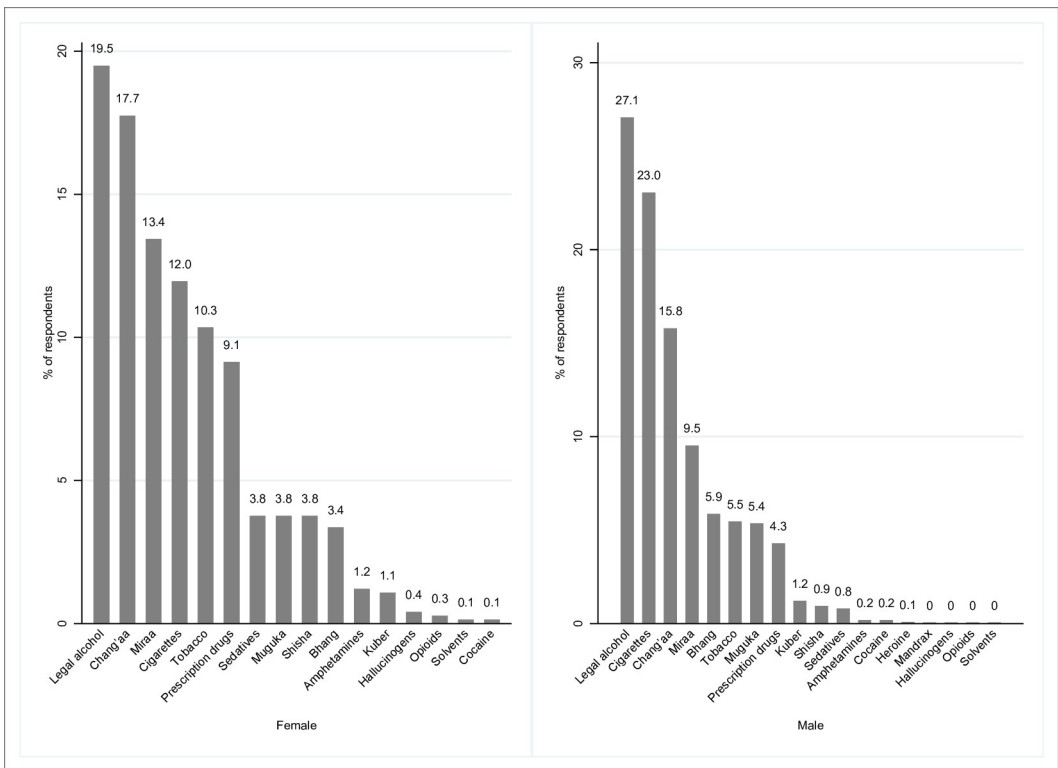

**Fig 5. Types of drugs and substances abused by the study participants grouped by gender.**

In Murang'a, the results from the qualitative surveys established that youth males are most affected by the abuse.

"*They abuse drugs so that they could gather courage to interact with others especially of the opposite sex.*" (IDI-current abuser)

Participants in the FGDs in Nyamira County reported that the problem cuts across all age groups and both genders. However, the FGDs participants reported that it is male youths who mostly abused drugs and substances. The other categories of individuals that were linked with drugs and substance abuse were the *bodaboda* riders, school dropouts and some old people.

"*Others are the tablets that mad or crazy people are given and most people use it even though they are not crazy*" (FGD- male youth)

Most of the respondents 41.4% (1,168) sourced their drugs from retail outlets (like kiosks, shops, kibanda, supermarkets), followed by entertainment establishments 24.0% (678) (like clubs, pubs, bars, restaurants), dealers in market/town 15.2% (430), home peddlers/local brewers 10.2% (288), and chemists/health centers 5.1% (144) (Fig 6). For each county, retail outlets were the most common sources of drugs for the respondents. Isiolo had the highest proportion of retail outlets selling drugs or substances at 46.3% (285), Kajiado 45.9% (275), Murang'a 42.1% (340) and Nyamira 33.6% (268).

In Isiolo County, results of the IDI with persons who had previously been abusing drugs and substances showed that the main sources of drugs and substances include friends, local shops and drug peddlers.

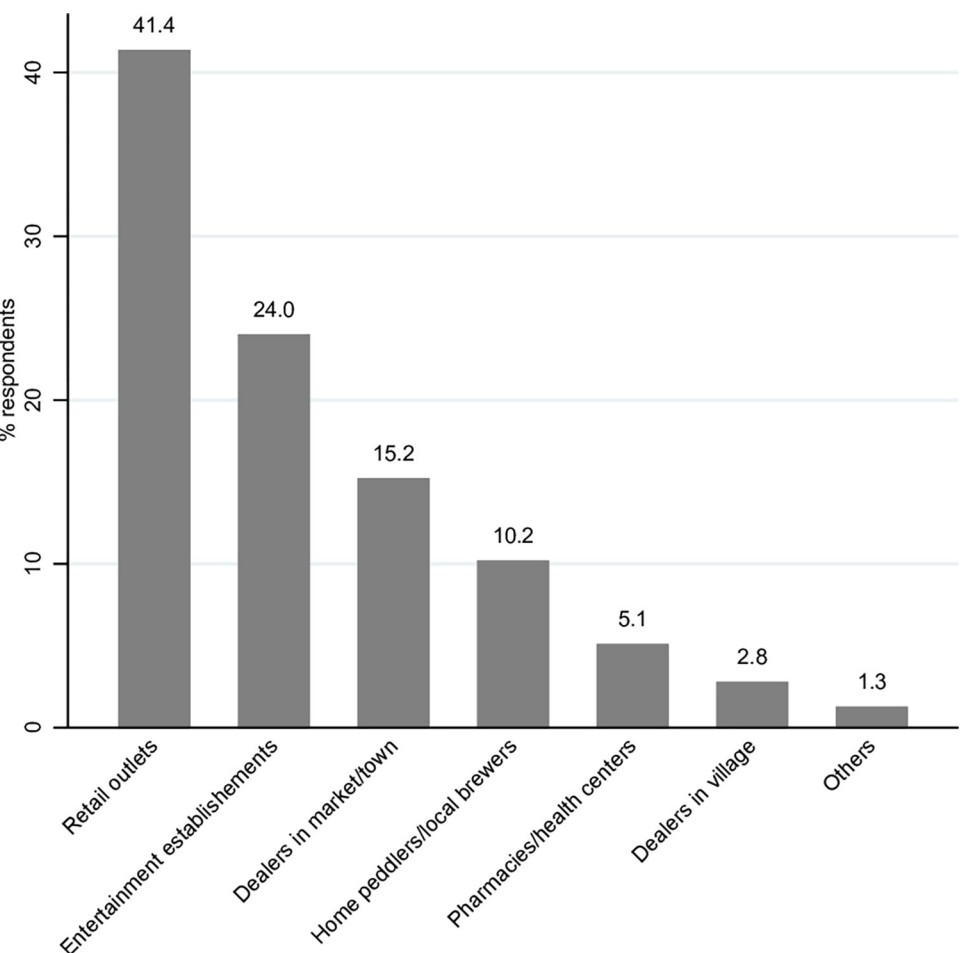

**Fig 6. The reported sources of drugs and substances abused by the participants.**

*"These drugs are all over because there are various sellers so it will depend on the type of drug you are using"* (IDI- Previous male drugs abuser).

*"Wallahi, there is a place in town where all the drugs you want are supplied we would send the young boys there to bring our orders"* (IDI-Previous female drugs abuser).

*"Drug peddlers, people would open shops or other businesses and in these, they would sell these drugs secretly. One would approach them as a normal buyer then purchase them. I never knew how to get them but friends taught me how to acquire them. After two to three years, I was an expert of who sells them, how to acquire them and their language"* (IDI-Previous male drugs abuser).

The results of IDIs with persons who had previously abused drugs and substance from Kajiado County showed that their drugs and substances are sourced from vendors/shops and friends.

*"Like this place you cannot miss drugs, it's everywhere, your friends can give you, I buy for myself."* (IDI-Previous drugs and substance abuser)

*"I was buying these drugs from the shop, they are in Bissil Town"* (IDI-Previous drugs and substance abuser)

*"These drugs you get just here in towns and the place you go you cannot miss them, like me normally I got from Kalito base like jaba that is miraa (khat). Bhang, I have been raised where bhang is being sold so I found myself at long last I have started using those things and I have gone 4 years using it."* (IDI- Previous drugs and substance abuser).

In Murang'a County, some of the reported sources of these drugs and substances were from vendors, shops, bars and peddlers.

*"I used to get cigarettes from shops, bhang from distributors and alcohol from alcohol brewers. Illicit brews were available at home and from other homes. Provided you have money, alcohol is always available."* (IDI-Previous drugs and substance abuser).

*"Miraa is chewed at every time but for the case of bhang they abuse it in hidden places. Any time they abuse bhang as long as there is nobody seeing them. Alcohol they take it in bars, in this village there is no place where they cannot get alcohol. They take it according to time they need it."* (IDI- Opinion leader).

*"Now like these kids smoke during day time and go to the bushes they take drugs and then take a walk, you will see them seated down like they are relaxing but they are smoking bhang."* (IDI- Opinion leader)

Results from Nyamira County showed that the main sources of drugs and substances were local vendors and own parents who were brewing alcohol. The frequency of abuse was in the evenings, during the weekend and when in the company of friends.

*"I used these drugs during weekends but when I became addicted I started consuming them regularly. Actually I got these drugs from clubs and bars."* (IDI- Previous drugs and substance abuser)

*"I used to start in the morning as soon as I woke up, I bought the drugs and friends also gave me, used to buy from local supermarkets and bars and sometimes from local chang'aa dens."* (IDI- Previous drugs and substance abuser)

*"We would buy them within the village and we go to hideouts and take them there."* (IDI-Previous drugs and substance abuser)

## Risk factors associated with drugs and substances abuse

From the univariable analysis, household heads from Nyamira County were more likely to abuse any drugs or substances (OR = 2.27, (95%CI: 1.73–2.97), p<0.001) as compared to household heads from Kajiado, Murang'a and Isiolo counties. Participants aged 45–53 years had greater odds of abusing any drugs or substances compared to those aged 18–26 years (OR = 1.27, (95%CI: 0.88–1.83), p = 0.001). Males had eight times greater odds of abusing any type of drugs or substances as compared to females (OR = 8.16, (95%CI: 6.56–10.15), p = 0.001). Marital status was also a key determinant of drugs and substances abuse with participants who were reportedly unmarried having greater odds of abuse (OR = 2.03, (95%CI: 1.36–3.02), p = 0.001) compared to those who were currently married. Participants with a secondary or college level of education had greater odds of abusing any drugs or substances (OR = 2.34, (95%CI: 1.51–3.62), p = 0.001) and (OR = 3.32, (95%CI: 1.89–5.80), p = 0.001) respectively compared to those with no formal education. Respondents who were government employees (OR = 4.93, (95%CI: 2.55–9.51), p = 0.001), students (OR = 3.89, (95%CI: 1.87–

8.13), p = 0.001), private worker (OR = 3.91, (95%CI: 1.95–7.83), p = 0.001), casual laborer (OR = 3.88, (95%CI: 2.73–5.53), p = 0.001), and self-employed (OR = 3.05, (95%CI: 2.17–4.28), p = 0.001) all had greater odds of abusing drugs and substances compared to those practicing subsistence farming (Table 4).

**Table 4. Univariable analysis of risk factors associated with any type of drugs and substances in Kenya.**

| Demographic characteristics | Univariable analysis |
|---|---|
| | OR (95%CI), p-value |
| County | |
| Kajiado | Reference |
| Murang'a | 1.51 (1.18–1.95), p = 0.001* |
| Nyamira | 2.27 (1.73–2.97), p = 0.001* |
| Isiolo | 1.91 (1.43–2.55), p = 0.001* |
| Age group (in years) | |
| 18–26 | Reference |
| 27–35 | 1.09 (0.79–1.50), p = 0.600 |
| 36–44 | 1.09 (0.77–1.56), p = 0.606 |
| 45–53 | 1.27 (0.88–1.83), p = 0.207 |
| 54–62 | 0.88 (0.60–1.27), p = 0.491 |
| 63–71 | 0.40 (0.28–0.58), p = 0.001* |
| ≥72 | 0.46 (0.29–0.71), p = 0.001* |
| Gender | |
| Male | 8.16 (6.56–10.15), p = 0.001* |
| Female | Reference |
| Marital status | |
| Single | 2.03 (1.36–3.02), p = 0.001* |
| Currently married | Reference |
| Divorced | 1.09 (0.68–1.76), p = 0.707 |
| Widow/widower | 0.35 (0.26–0.47), p = 0.001 |
| Education level | |
| None | Reference |
| Primary | 1.11 (0.74–1.68), p = 0.604 |
| Secondary | 2.34 (1.51–3.62), p = 0.001* |
| College | 3.32 (1.89–5.80), p = 0.001* |
| Occupation | |
| Subsistence farmer | Reference |
| Government employee | 4.93 (2.55–9.51), p = 0.001* |
| Student | 3.89 (1.87–8.13), p = 0.001* |
| Large scale farmer | 1.47 (0.32–6.67), p = 0.621 |
| Private worker | 3.91 (1.95–7.83), p = 0.001* |
| Casual laborer | 3.88 (2.73–5.53), p = 0.001* |
| Self employed | 3.05 (2.17–4.28), p = 0.001* |
| Others | 0.82 (0.52–1.29), p = 0.391 |
| Dwelling | |
| Permanent | 1.27 (0.99–1.63), p = 0.064 |
| Semi-permanent | Reference |
| Temporary | 1.63 (1.14–2.32), p = 0.007* |

*Indicates a significant risk factor during the univariable analysis

**Table 5. Multivariable analysis of risk factors associated with any type of drugs and substances in Kenya.**

| Demographic characteristics | Multivariable analysis |
| --- | --- |
|  | (aOR (95%CI), p-value) |
| Age group (in years) |  |
| 18–26 | Reference |
| 27–35 | 1.56 (0.96–2.54), p = 0.071 |
| 36–44 | 1.19 (0.70–2.02), p = 0.499 |
| 45–53 | 1.68 (0.97–2.90), p = 0.063 |
| 54–62 | 1.01 (0.58–1.76), p = 0.975 |
| 63–71 | 0.59 (0.32–1.06), p = 0.078 |
| ≥72 | 0.49 (0.25–0.95), p = 0.035 |
| Gender |  |
| Male | 7.02 (5.21–9.45), p = 0.001** |
| Female | Reference |
| Marital status |  |
| Single | 1.71 (1.06–2.77), p = 0.028** |
| Currently married | Reference |
| Divorced | 0.97 (0.58–1.62), p = 0.722 |
| Widow/widower | 1.08 (0.74–1.58), p = 0.683 |
| Education level |  |
| None | Reference |
| Primary | 0.42 (0.26–0.69), p = 0.001 |
| Secondary | 0.53 (0.31–0.92), p = 0.024 |
| College | 0.67 (0.33–1.38), p = 0.277 |
| Occupation |  |
| Subsistence farmer | Reference |
| Government employee | 2.27 (1.05–4.91), p = 0.036** |
| Student | 2.23 (0.86–5.74), p = 0.097 |
| Large scale farmer | 1.04 (0.19–5.55), p = 0.963 |
| Private worker | 1.95 (0.91–4.17), p = 0.085 |
| Casual laborer | 1.99 (1.32–2.98), p = 0.001** |
| Self employed | 1.86 (1.24–2.77), p = 0.003** |
| Others | 0.89 (0.52–1.53), p = 0.687 |

**Indicates a significant risk factor during the multivariable analysis

Multivariable analysis revealed that drugs and substances abuse was significantly higher among males (aOR = 7.02, (95%CI: 5.21–9.45), p = 0.001), government employees (aOR = 2.27, (95%CI: 1.05–4.91), p = 0.036) and unmarried participants (aOR = 1.71, (95%CI: 1.06–2.77), p = 0.028) (Table 5).

Results of the IDIs with current drug abusers, persons who had previously abused drugs and substances, health workers and FGD participants from Isiolo county gave causes for drugs and substance abuse that included: to be able to interact with others, for fun and to relax, to cope with stress, relate freely with opposite sex, lack of employment, poverty and peer pressure, drugs being readily available and lack of parental guidance and mentorship and influence from the social media.

"*Mostly I use them for leisure, to pass time and to feel good*" (IDI-Current adult drugs abuser).

*"My friends used to chew miraa smoke bhang and take chang'aa. They persuaded me to taste and I gave it a try that's how I ended up taking drugs. I saw it as something good"* (IDI- Current mature minor drugs abuser).

*"When my mother passed away since I had stress and the family that we formed there was stress due to the issues brought by the step mother. At that time I joined a company of other friends who encouraged me to smoke bhang to relieve stress. I used and came to realize that I became drunk and the stress was no more and I could sleep well"*(IDI-Current mature minor drugs abuser).

In Kajiado County, the IDIs established that most drugs and substances were abused in the forests of Emotoroki location where charcoal burning is common. The abuse also takes place along the river banks of Majengo Township and during traditional ceremonies and school holidays by school children.

*"Their usage according to your question is mostly during their activities of charcoal burning and I don't know if they think it will add them more strength in their work."* (IDI- Community Opinion Leader)

*"Beers and other packed alcohols are taken in bars while brews like Chang'aa and those others are taken in their places of brewing. Miraa and Bhang are taken in the hidden places. Those who take them know all these places and that's where they stay while taking them. They are mostly located in the bush or in small structures where there is nobody."* (IDI- Community Opinion Leader)

For Murang'a County, in-depth interview results with current drugs and substance abusers further revealed some of the reasons for abuse include coping with stress, for fun, peer pressure and idleness while persons who had previously abused drugs and substances mentioned stress, peer pressure, idleness and being brought up by abusive parents as reasons for their having been drug abusers.

"*And I used because when I started I had my friends, you know when I finished, I had not gotten. . .I had no job it is that you are that idle so you get there is that peer pressure, they come we chat, we go to certain places together. For example, me I lost my parents early, so there's that stress I got and remember many things, so it's that I just say let me use some other stress to be over.*" (IDI- Previous drug abuser)

*"There is peer influence, changing social ways of life, like children are able to access cash anyway . . . Aaah. . . there is also misuse of leisure which may lead them into spending their leisure in a place which will introduce these things to clubbing. Aaaa. . . . . .. I can also talk about changes in the community. Like even to some extent poverty. . ..eeeeh because of. . . .why I tend to think that poverty can lead to them to this or hardship in lifestyle is getting away this boys and girls gets frustrated along the way. They get stressed. If their needs are not met and as they are searching for their needs they find themselves being introduced to things such as these drugs as a way to suppress their mind.*" (IDI- Current drug abuser, adult)

In Kajiado County the results from the IDIs with health workers further revealed that some of the reasons attributable to drugs and substance abuse among male youth included communities not prioritizing education and children being sent to herd cattle, lack of parental supervision, during circumcision when youth are left to stay away from home for some time, peer pressure and idleness. Female youth were reported to have no time to idle around as they would be expected to perform house chores.

*"Then even the culture. They say, stay together at a certain age, I think at the age of circumcision, youth being in a room for a month, you don't know what they do, so even the culture gets them in those things."* (IDI-Health worker)

*"Education you can find an educated person engaging into drug and substance abuse very few in a small portion, so you find that most of the drug abusers are in slums. Why? Because most of them are idle, so from the idling they engage into drugs. The other thing is peer pressure, you find your friend is using so you say that let me also try. And also if we have a steady supply of drug more people will tend to use it, because there is availability unless you cut the supply, the user will be very low.'* (IDI- Health worker)

*"Youth male, due to low socio-economic status."* (IDI-Health worker)

*"You know the men mostly do not have work to do but the female gender they are preoccupied with work to do especially the ladies are the ones who have to look after the goats and animals, fetch water at far distances also get firewood and many other things in the homestead that mostly the woman do compared to men where by most of them just leave in the morning and come back in the evening."* (IDI-Health worker).

Results from the FGDs with community group in Nyamira County also indicated that causes for drug and substances reported include peer influence, availability of the substances and drugs, unemployment of the youth, urbanization, poverty and lack of parental care.

## Discussion

In Kenya, several studies have highlighted the serious nature of substance use [5–7]. This study generated data on the magnitude, types, patterns, sources and risk factors of drugs and substances abused. This is in line with NACADA policies on use of evidence-based approaches in the fight against alcohol, drug and substance abuse which calls for research on diverse aspects within the realm of substance use and abuse [8]. The data from this study will be useful in guiding the development, and implementation of policies that would address drug abuse which if left unaddressed could lead to lost generations.

The overall prevalence of DSA was high (> 80%) compared to a previous national survey by NACADA in 2012 which indicated a national prevalence of 37.1% [8]. The findings that drug abuse is more prevalent among males are consistent with other studies in Kenya [7, 8]. Magnitude of any drugs and substances abuse decreased with age and this finding has emphasized that abuse is highest in populations within the ages of 15 to 65 years [14].

In terms of types of drugs used, findings are consistent with other studies in Kenya that revealed alcohol, cigarettes, and chang'aa comprising both illicit and licit are the substances that are widely abused in Kenya [15–18]. Also, similar to a study done among people with disabilities (PWDs) which revealed alcoholic beverages as the most abused, followed by tobacco products, khat/miraa and marijuana [19]. Our findings showed that legal/packaged alcohol was the most abused type of drugs/substances in both Kajiado and Murang'a while chang'aa was most abused in Nyamira. This corroborated the findings from a study conducted in 2017 that showed alcohol as the widely abused substance in Nairobi, Eastern, Western and Rift Valley regions [14]. In Isiolo, the findings showed that khat/miraa was the most abused type of drug, a result that is supported by a previous study done in Eastern region in 2017 that showed prevalence of 8.5% for khat/miraa use in the region [14]. The fact that alcoholic beverages, tobacco products and khat/miraa were the widely abused substances by the participants could be because of their ease of accessibility and affordability compared to the hard drugs [20]. The

study showed that the leading source of drugs and substances for abusers was retail outlets, a finding that was similarly pointed out by Kathungu and colleagues [19].

The observed high prevalence of drug use among teens and youths (<24 years) was a reflection on the need to address problems such as unemployment, neglect, violence, sexual abuse and poor academic performance, that may be driving them to engage in the vice [8]. Specifically, among the youths, this issue is also reported as a cause of problems in learning institutions in Kenya [5, 6]. In the study, a higher educational level was not a protective factor against drugs and substances abuse. Participants who reported their marital status as single had significantly greater odds of drugs and substances abuse compared to other types of marital status (married or widowed). This is not unusual considering some of the household heads abusing drugs and substances were <36 years (36.5%) and majority of the younger populations tend to have their marital status as single. Across the types of occupation, the fact that government employees were twice more likely to abuse drugs and substances maybe attributed to having a stable income and therefore an increased ability to buy or obtain the drugs and substances of abuse.

## Study strengths and limitations

First, the study used purposive sampling technique to sample counties, sub-counties, locations, and sub-locations. Whereas, this technique increased the results credibility, transferability, dependability, and confirmability, it however reduced the ability of the study to generalize the results since it is subject to researcher and sampling bias [21]. Second, the use of systematic sampling of the households (and household heads) provided a probabilistic approach to this mixed methods study, hence enabling us to overcome the first limitation above. Third, the study targeted households where there is at least one user of drugs or substances. This household inclusion criterion introduced some bias to the sample and potentially inflated the calculated prevalence. Fourth, the mixed method approach used in this study enhanced the robustness of the reported results and rigorously poised this study to influence policy on management of the DSA situation in the studied counties. Lastly, the study did not stratify the data collection by gender and age groups, hence limited the study in conclusively detecting the differences in drug or substance use by these important demographics.

## Conclusions

Results of this study have shown a high prevalence of drugs and substances abuse among the participants compared to previous national surveys. This growing nature and worrying trend of the abuse calls for more effective measures to prevent and manage the problem. The overall study findings generated knowledge that will be useful in development of specific control programmes necessary to address age, gender and county needs of drugs and substances abuse in Kenya in order to respond to the problem comprehensively. This study was conducted in line with NACADA's mandate to promote use of research on drugs and substances abuse to guide development of policy and programmes that meet the needs of the specific populations.

## Acknowledgments

We are grateful to the respective Counties Ministries of Health, the respective Counties Commissioners and their Assistant County Commissioners. Special thanks go to all members of the study team, field personnel and the participants for their commitment towards this work. Heartfelt thank you to the entire team at *Colozzy Data Analytics and Research Solutions* led by Abigael Wangari, Evans Kosgei, Lilian Owino and Melvine Madoro for helping with data

cleaning and management. This paper is published with the permission of the Director General, KEMRI.

## Author Contributions

**Conceptualization:** Collins Okoyo, Elizabeth Echoka, Charles Mwandawiro, Doris Njomo.

**Data curation:** Collins Okoyo, Vincent Were, Sylvie Araka, Henry Kanyi, Elizabeth Echoka, Charles Mwandawiro, Doris Njomo.

**Formal analysis:** Collins Okoyo, Elizabeth Njambi, Doris Njomo.

**Funding acquisition:** Doris Njomo.

**Investigation:** Collins Okoyo, Henry Kanyi, Linnet Ongeri, Elizabeth Echoka, Charles Mwandawiro, Doris Njomo.

**Methodology:** Collins Okoyo, Vincent Were, Henry Kanyi, Linnet Ongeri, Elizabeth Echoka, Charles Mwandawiro, Doris Njomo.

**Project administration:** Doris Njomo.

**Supervision:** Collins Okoyo, Vincent Were, Sylvie Araka, Henry Kanyi, Linnet Ongeri, Elizabeth Echoka, Charles Mwandawiro, Doris Njomo.

**Validation:** Collins Okoyo.

**Visualization:** Collins Okoyo.

**Writing – original draft:** Collins Okoyo, Elizabeth Njambi, Doris Njomo.

**Writing – review & editing:** Collins Okoyo, Elizabeth Njambi, Vincent Were, Sylvie Araka, Henry Kanyi, Linnet Ongeri, Elizabeth Echoka, Charles Mwandawiro, Doris Njomo.

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
