## [Decision Letter · Decision Letter 0]

10 Feb 2022

PONE-D-22-01584Types, patterns and risk factors associated with drugs and substances abuse: A cross-sectional study of selected counties in KenyaPLOS ONE

Dear Dr. Okoyo,

Thank you for submitting your manuscript to PLOS ONE. After careful consideration, we feel that it has merit but does not fully meet PLOS ONE’s publication criteria as it currently stands. Therefore, we invite you to submit a revised version of the manuscript that addresses the points raised during the review process.

Based on my opinion and those of the reviewers, although the manuscript presents important theme and information it might not be interest to readers without reworking on the methodology and proper analysis of the the data. I advise that the paper be re-written considered all the issues raised by the reviewers. At this stage I can state that its acceptance for publication will depend on the outcome of the revision.==============================

We look forward to receiving your revised manuscript.

Kind regards,

Gabriel O Dida, PhD

Academic Editor

PLOS ONE

Journal Requirements:

"we want to thank the Government of Kenya through KEMRI for funding this work. "

"The financial support for this research was provided by the Government of Kenya through KEMRI/GRG/15/31. The funders had no role in the study design, data collection and analysis, decision to publish or preparation of the manuscript."

Reviewers' comments:

Reviewer's Responses to Questions

**Comments to the Author**

1. Is the manuscript technically sound, and do the data support the conclusions?

Reviewer #1: No

Reviewer #2: Partly

Reviewer #3: Yes

Reviewer #4: No

2. Has the statistical analysis been performed appropriately and rigorously? 

Reviewer #1: No

Reviewer #2: Yes

Reviewer #3: Yes

Reviewer #4: No

3. Have the authors made all data underlying the findings in their manuscript fully available?

Reviewer #1: Yes

Reviewer #2: No

Reviewer #3: Yes

Reviewer #4: No

4. Is the manuscript presented in an intelligible fashion and written in standard English?

Reviewer #1: No

Reviewer #2: No

Reviewer #3: Yes

Reviewer #4: Yes

5. Review Comments to the Author

Reviewer #1: Dear Authors,

I have read the manuscript and I think that it has an importan epidemiological issue, but I think that it must be rewritten

1) Title: must be revised respect to the data presented

2) Introduction: it must be rewritten considering the title: drug use abnd abuse so other considerations must be deleted

3) Methods: Experimental protocol is missing

4) Results: these are hard to understand I thin that must be completely rewrittex and as reported in results must be enclosed in a supplement file. In results you must add the data enclosed in tables

Reviewer #2: I have carefully reviewed this manuscript (ms.) which seems interesting and focused on a topic of international interest such as the use and abuse of legal and illegal drugs.

This ms. focuses on this research topic in Kenya

Although the work is dense and contains a lot of information, it presents some theoretical and methodological limitations. However, globally, this ms. allows me to conclude that it could be accepted for publication in PLOS ONE with minor changes.

This ms. needs English refinement.

ABSTRACT

Indicate the type of sampling conducted in this study.

Indicate, at least, the total N, % men or women, age range or age groups, mean and SD.

The information regarding the type of sampling and characteristics of the samples according to Figure 2 is not clear. Everything is ambiguous and imprecise.

The samplings conducted in this ms. they are non-probabilistic or for convenience. This is the most important limitation of this ms. while this poses a threat to the internal, but above all external, validity of the results found in this study. This concern could be reason enough for this ms. was rejected for publication in PLOS ONE.

INTRODUCTION

This section is properly focused and centered. The authors review previous empirical evidence found in Kenya. This is correct.

However, the authors should robustly justify the novelty, improvement or scientific advance of this ms. compared to previously published studies in Kenya. This concern is key. Authors should include this information clearly and precisely.

The authors end this section by indicating the general objective of this study. However, the authors should clearly and precisely indicate the specific objectives of this ms., as well as the hypotheses corresponding to each specific objective. Of course, these hypotheses must be formulated based on the previous empirical evidence found in Kenya, not on the results found in this study. Furthermore, these hypotheses should be robustly accepted or rejected in the Discussion section.

PARTICIPANTS

Bearing in mind Figure 2, authors should provide data for all categories or levels presented in this figure the following data: N, % male or female, age range, mean and SD). Everything is confusing.

How were outliers and missing data handled statistically?

Reviewer #3: The current manuscript describes drug and substance abuse in four counties in Kenya. The authors collected quantitative and qualitative data assessing the impact of age, educational level, gender, etc., on use and abuse of different substances in four counties and sub-counties. The authors report that there is an increase in drug and substance use in the present study compared to the one conducted in 2012. In addition, they reported that males consume drugs and substances more than females, yet the level of education did not matter. Age was another factor, showing younger adults and elderly (65 and youngers) consume more than older than 65. Overall, the manuscript contains useful information in terms of management of addiction and policy making strategies in Kenya and particularly in those counties. However, the manuscript is too lengthy and there are some information that may be deleted unless the authors need to discuss them and provide a rationale why such information is necessary to be included in the manuscript.

Major:

1. The authors state collected data but not analyzed whether the religion of the subjects impacted drug and substance abuse. If there is enough power in the study, this should analyzed and discussed.

2. Some information can be deleted or can be shortened. For example, three sentences on lines 154-158 and four sentences on lines 163-168, lines 172-178), and lines 182-186. Instead, please indicate why these counties are purposively selected since it is not obvious until later. This information also should be included in the Abstract as to why those counties were selected.

3. The authors discussed the types of drugs and concluded that heavy drugs were used less than alcohol and tobacco (line 332-335). Was this due to the cost of heavy drugs or their lack of availability as compared to alcohol? This should be discussed.

4. I think the information regarding the interviews provided in the result section can be presented in a supplemental data since this information makes the current manuscript too lengthy. However, I leave this to the discretion of the editor.

5. The number of tables and figures can be reduced. Some information is redundant.

Minor:

1. Please include other nicotine products next to cigarettes (line 83).

2. Please change psycho-stimulants to psychostimulants (Line 87).

3. Please delete the semicolon on line 277 and 279 or replace them with colon (:).

4. Please add "and" before "village elder" on line 280.

5. Please remove the comma after majority on line 302.

Reviewer #4: Thank you for your hard work dong all these types of data collection and analysis. However, I have some comments:

1. In your methodology for the quantitative part of the study the selection of the participants was not clear. You mentioned in page 5 line 148 that you used a systematic sampling technique without describing how you did that. But you mentioned that you will take houses when one at least of the household is using drugs, this is a purposive sample. We know that when one of the household is using drugs (as you also included smoking as one of these drugs) mostly more than one will be using drugs as well. So you could not calculate prevalence from this sample as this will be a biased sample.

2. On the other hand, in tables 2 and 3 you are presenting the types and frequency of drug used per age groups and per county. In the last row you are putting a total and %. You did not consider the multiple drug users in these tables and so the percent you are calculating is not a real percent.

3. You also did not mention how you asked your question to let us know if the smokers and other drug users are ever users or current users and what type of questionnaire you used. Is your questionnaire validated in the participants language or not?

4. At the end you concluded that the drug use is high with a purposive sample and without considering the multiple drug users, I could not accept your conclusion as it built on biased data.

5. Regarding the references, you used only 17 references, one of the them World Drug Report 2018 and another one was the WHO global status report 2011, and only 15 per reviewed, although in this area many publications are there.

6. Finally regarding the maps, it would be better as well to let us know where are the counties in the country.

6. PLOS authors have the option to publish the peer review history of their article (what does this mean?). If published, this will include your full peer review and any attached files.

Reviewer #1: No

Reviewer #2: **Yes: **Candido J. Ingles

Reviewer #3: **Yes: **Kabirullah Lutfy

Reviewer #4: No

---

## [Author Response · Author response to Decision Letter 0]

18 May 2022

PONE-D-22-01584

Types, patterns and risk factors associated with drugs and substances abuse: A cross-sectional study of selected counties in Kenya

PLOS ONE

Academic Editor Comments:

Based on my opinion and those of the reviewers, although the manuscript presents important theme and information it might not be interest to readers without reworking on the methodology and proper analysis of the the data. I advise that the paper be re-written considered all the issues raised by the reviewers. At this stage I can state that its acceptance for publication will depend on the outcome of the revision.

>> We have endeavoured to revise this manuscript based on your assessment and that of the reviewers.

"The financial support for this research was provided by the Government of Kenya through KEMRI/GRG/15/31. The funders had no role in the study design, data collection and analysis, decision to publish or preparation of the manuscript."

>> Please use this statement as the funding statement.

>> We have moved the ethics statement to the appropriate section

>> We created the figure from scratch using the ArcGIS software, hence it is not copyrighted.

 >> We created figure 1 from scratch using the ArcGIS software, hence it is not copyrighted.

>> We created figure 1 from scratch using the ArcGIS software, hence it is not copyrighted.

Reviewers Comments to the Author

Reviewer #1: Dear Authors,

I have read the manuscript and I think that it has an importan epidemiological issue, but I think that it must be rewritten

>>Thanks for this critical assessment of our paper. We have addressed each of the concerns of the reviewers.

1) Title: must be revised respect to the data presented

>> We have rephrased the title appropriately.

2) Introduction: it must be rewritten considering the title: drug use abnd abuse so other considerations must be deleted

>> Thanks for this comment, we have revised this section based on specific appropriate comments.

3) Methods: Experimental protocol is missing

>> This was a descriptive study with no experimentation performed or intervention given.

4) Results: these are hard to understand I thin that must be completely rewrittex and as reported in results must be enclosed in a supplement file. In results you must add the data enclosed in tables

>> Thanks for this comment. We have reviewed again this section with the aim to improve its readability. All data reported in this section are detailed in the appropriate tables. Remember, we are reporting a mixed methods results, with excerpts from the qualitative arm.

Reviewer #2: I have carefully reviewed this manuscript (ms.) which seems interesting and focused on a topic of international interest such as the use and abuse of legal and illegal drugs.

This ms. focuses on this research topic in Kenya

Although the work is dense and contains a lot of information, it presents some theoretical and methodological limitations. However, globally, this ms. allows me to conclude that it could be accepted for publication in PLOS ONE with minor changes.

>> Thanks for your complimenting assessment of our work. We have endeavoured to improve it based on the reviewers’ comments and editor assessment. Furthermore, we have detailed some of the study strengths and limitations, lines 732-744.

This ms. needs English refinement.

>> We have gone through the entire manuscript and corrected all the typos. It reads better now.

ABSTRACT

Indicate the type of sampling conducted in this study.

>> Under methods subsection, we have indicated that both purposive and systematic sampling were used where appropriate.

Indicate, at least, the total N, % men or women, age range or age groups, mean and SD.

>> Please see lines 35 – 43 where we have indicated the total sample size (N=3600). Due to word limitation in this section, other details like age ranges, mean and SD are left in the main results section.

The information regarding the type of sampling and characteristics of the samples according to Figure 2 is not clear. Everything is ambiguous and imprecise.

>> This figure is self-explanatory. Please specify what is unclear and/or what should be improved. The figure shows the hierarchy of data collection. The details of sampling per stage is in the main text.

The samplings conducted in this ms. they are non-probabilistic or for convenience. This is the most important limitation of this ms. while this poses a threat to the internal, but above all external, validity of the results found in this study. This concern could be reason enough for this ms. was rejected for publication in PLOS ONE.

>> Thanks for your comment, whilst we agree with you that the mostly used sampling technique in most stages in this study was purposive. However, for the quantitative arm, households (and by extension participants) were sampled using a systematic sampling technique, which is a probabilistic approach, thereby giving the study some validity and generalizability. Additionally, this study used a mixed method design and therefore qualitative components doesn’t necessarily require probabilistic sampling approaches. Additionally, we have acknowledged the limitation in the use of purposive sampling (lines 732-744), and provided justification for its use (lines 152-157).

INTRODUCTION

This section is properly focused and centered. The authors review previous empirical evidence found in Kenya. This is correct.

>> Thanks for this compliment. We appreciate

However, the authors should robustly justify the novelty, improvement or scientific advance of this ms. compared to previously published studies in Kenya. This concern is key. Authors should include this information clearly and precisely.

>> The closing introductory paragraph, lines 103-120, already highlighted the general problems with DSA, government efforts in controlling the problem, and how this study complement these efforts.

The authors end this section by indicating the general objective of this study. However, the authors should clearly and precisely indicate the specific objectives of this ms., as well as the hypotheses corresponding to each specific objective. Of course, these hypotheses must be formulated based on the previous empirical evidence found in Kenya, not on the results found in this study. Furthermore, these hypotheses should be robustly accepted or rejected in the Discussion section.

>> Thanks for this comment. Please note that the specific objectives are nuanced within the overriding objective as stated in lines 116-120. About the hypotheses, please note that this was a descriptive study and therefore need not necessarily have testable hypotheses.

PARTICIPANTS

Bearing in mind Figure 2, authors should provide data for all categories or levels presented in this figure the following data: N, % male or female, age range, mean and SD). Everything is confusing.

How were outliers and missing data handled statistically? 

>> Please this comment is not clear. The figure was only meant to show the hierarchical nature of the collected data and various stages of sampling. The details you are requesting are already in the methods and results sections, and were not meant to be shown in this figure.

Reviewer #3: The current manuscript describes drug and substance abuse in four counties in Kenya. The authors collected quantitative and qualitative data assessing the impact of age, educational level, gender, etc., on use and abuse of different substances in four counties and sub-counties. The authors report that there is an increase in drug and substance use in the present study compared to the one conducted in 2012. In addition, they reported that males consume drugs and substances more than females, yet the level of education did not matter. Age was another factor, showing younger adults and elderly (65 and youngers) consume more than older than 65. Overall, the manuscript contains useful information in terms of management of addiction and policy making strategies in Kenya and particularly in those counties. 

>> Thanks for this accurate appraisal of our manuscript.

However, the manuscript is too lengthy and there are some information that may be deleted unless the authors need to discuss them and provide a rationale why such information is necessary to be included in the manuscript.

>> Thanks for this suggestion, we have endeavoured to improve the document based on comments from both the reviewers and the editor.

Major:

1. The authors state collected data but not analyzed whether the religion of the subjects impacted drug and substance abuse. If there is enough power in the study, this should analyzed and discussed.

>> Thanks for the comment. However, the data collection was not stratified by the religion status hence we could not firmly analyze data by religion. Actually, the observations were imbalanced between the religion status, even the regression analysis performed did not show any significant differences between christians and muslims in terms of drug and substances abuse. 

2. Some information can be deleted or can be shortened. For example, three sentences on lines 154-158 and four sentences on lines 163-168, lines 172-178), and lines 182-186. Instead, please indicate why these counties are purposively selected since it is not obvious until later. This information also should be included in the Abstract as to why those counties were selected.

>> We have shortened some details in this section and also provided in the abstract why counties were selected purposively.

3. The authors discussed the types of drugs and concluded that heavy drugs were used less than alcohol and tobacco (line 332-335). Was this due to the cost of heavy drugs or their lack of availability as compared to alcohol? This should be discussed.

>> We have added a statement about this in the discussion, see lines 716-718.

4. I think the information regarding the interviews provided in the result section can be presented in a supplemental data since this information makes the current manuscript too lengthy. However, I leave this to the discretion of the editor.

>> Thanks for this suggestion. However, these qualitative information help in enhancing the understanding of the reader. Moreover, we have deleted, throughout the manuscript, areas where we think there were redundancy.

5. The number of tables and figures can be reduced. Some information is redundant.

>> There are only 5 tables and 6 figures, with all providing useful information to the readers.

Minor:

1. Please include other nicotine products next to cigarettes (line 83).

>> We have included this.

2. Please change psycho-stimulants to psychostimulants (Line 87).

>> Done

3. Please delete the semicolon on line 277 and 279 or replace them with colon (:).

>> Done

4. Please add "and" before "village elder" on line 280.

>> Done

5. Please remove the comma after majority on line 302.

>> Done

Reviewer #4: Thank you for your hard work dong all these types of data collection and analysis. 

>> Thanks for this compliment. We appreciate

However, I have some comments:

1. In your methodology for the quantitative part of the study the selection of the participants was not clear. You mentioned in page 5 line 148 that you used a systematic sampling technique without describing how you did that. 

>> We have now explained how this systematic sampling was implement, please see lines 161-164.

But you mentioned that you will take houses when one at least of the household is using drugs, this is a purposive sample. 

>> This is not purposive sampling per se, but it is an inclusion criterion for the household selection.

We know that when one of the household is using drugs (as you also included smoking as one of these drugs) mostly more than one will be using drugs as well. So you could not calculate prevalence from this sample as this will be a biased sample.

>> It is true that this set criterion biased our sample to some extent and therefore limiting our calculation of “prevalence”. However, prevalence calculation was only based on the current users and not past users. We have acknowledged this limitation in the discussion, lines 732-744.

2. On the other hand, in tables 2 and 3 you are presenting the types and frequency of drug used per age groups and per county. In the last row you are putting a total and %. You did not consider the multiple drug users in these tables and so the percent you are calculating is not a real percent.

>> Thanks for pointing this out, we agree that this total percent could be misleading. We have deleted appropriately.

3. You also did not mention how you asked your question to let us know if the smokers and other drug users are ever users or current users and what type of questionnaire you used. Is your questionnaire validated in the participants language or not?

>> We collected information from various categories of participants using both quantitative and qualitative methods. Questionnaire was administered to the household heads who answered the questions on behalf of the entire household members, IDIs were conducted with various stakeholders and opinion leaders, and FGDs were conducted with separate groups including different groups for current users and past users. All data collection tools were validated and translated in participants local language.

4. At the end you concluded that the drug use is high with a purposive sample and without considering the multiple drug users, I could not accept your conclusion as it built on biased data.

>> The issue around purposive sampling has been explained in the text and the primary sampling units were selected using systematic sampling technique (a probabilistic approach). Again, the prevalence was for any type of drug or substance abused, hence multiple drug use cannot be a limitation in arriving at the conclusion.

5. Regarding the references, you used only 17 references, one of the them World Drug Report 2018 and another one was the WHO global status report 2011, and only 15 per reviewed, although in this area many publications are there.

>> There are no requirements on the number of references to be included in a study. The references we used were relevant to our study in terms of the population of interest, the geography, the time-frame of the studies among other aspects that the study team considered. Nonetheless, we have added a few references throughout the document.

6. Finally regarding the maps, it would be better as well to let us know where are the counties in the country.

>> Already the overview panel on the right hand side of the map show where the counties are in the country.

---

## [Decision Letter · Decision Letter 1]

10 Aug 2022

Prevalence, types, patterns and risk factors associated with drugs and substances of use and abuse: A cross-sectional study of selected counties in Kenya

PONE-D-22-01584R1

Dear Dr. Okoyo,

We’re pleased to inform you that your manuscript has been judged scientifically suitable for publication and will be formally accepted for publication once it meets all outstanding technical requirements.

Kind regards,

Gabriel O Dida, PhD

Academic Editor

PLOS ONE

Additional Editor Comments (optional):

Reviewers' comments:

Reviewer's Responses to Questions

**Comments to the Author**

1. If the authors have adequately addressed your comments raised in a previous round of review and you feel that this manuscript is now acceptable for publication, you may indicate that here to bypass the “Comments to the Author” section, enter your conflict of interest statement in the “Confidential to Editor” section, and submit your "Accept" recommendation.

Reviewer #2: All comments have been addressed

Reviewer #3: All comments have been addressed

2. Is the manuscript technically sound, and do the data support the conclusions?

Reviewer #2: No

Reviewer #3: Yes

3. Has the statistical analysis been performed appropriately and rigorously? 

Reviewer #2: No

Reviewer #3: Yes

4. Have the authors made all data underlying the findings in their manuscript fully available?

Reviewer #2: No

Reviewer #3: Yes

5. Is the manuscript presented in an intelligible fashion and written in standard English?

Reviewer #2: No

Reviewer #3: Yes

6. Review Comments to the Author

Reviewer #2: The authors' answers are not convincing, rigorous. Responses provided for the authors are few precise. This is not possible. The responses to reviewers are not appropiate. A convenience sampling not is possible in PLOS ONE.

Reviewer #3: The authors adequately responded to my comments of the previous version of the manuscript. I have not further comments or concerns.

7. PLOS authors have the option to publish the peer review history of their article (what does this mean?). If published, this will include your full peer review and any attached files.

Reviewer #2: **Yes: **Candido J. Ingles

Reviewer #3: **Yes: **Kabirullah Lutfy

---

## [Editor Report · Acceptance letter]

2 Sep 2022

PONE-D-22-01584R1 

Prevalence, types, patterns and risk factors associated with drugs and substances of use and abuse: A cross-sectional study of selected counties in Kenya 

Dear Dr. Okoyo:

I'm pleased to inform you that your manuscript has been deemed suitable for publication in PLOS ONE. Congratulations! Your manuscript is now with our production department. 

Kind regards, 

on behalf of

Dr. Gabriel O Dida 

Academic Editor

PLOS ONE